# Kalirin Interacts with TRAPP and Regulates Rab11 and Endosomal Recycling

**DOI:** 10.3390/cells9051132

**Published:** 2020-05-04

**Authors:** Xiaolong Wang, Meiqian Weng, Yuting Ke, Ellen Sapp, Marian DiFiglia, Xueyi Li

**Affiliations:** 1School of Pharmacy, Shanghai Jiao Tong University, Shanghai 200240, China; xiaolong29job@sjtu.edu.cn (X.W.); devillua@sjtu.edu.cn (Y.K.); 2Mucosal Immunology Lab combined program in Pediatric Gastroenterology and Nutrition, Massachusetts General Hospital and Harvard Medical School, Charlestown, MA 02129, USA; mweng@mgh.harvard.edu; 3Department of Neurology, Massachusetts General Hospital and Harvard Medical School, Charlestown, MA 02129, USA; esapp@mgh.harvard.edu (E.S.); difiglia@helix.mgh.harvard.edu (M.D.)

**Keywords:** guanine nucleotide exchange factor, Rab11, Rho, TRAPP, kalirin

## Abstract

Coordinated actions of Rab and Rho are necessary for numerous essential cellular processes ranging from vesicle budding to whole cell movement. How Rab and Rho are choreographed is poorly understood. Here, we report a protein complex comprised of kalirin, a Rho guanine nucleotide exchange factor (GEF) activating Rac1, and RabGEF transport protein particle (TRAPP). Kalirin was identified in a mass spectrometry analysis of proteins precipitated by trappc4 and detected on membranous organelles containing trappc4. Acute knockdown of kalirin did not affect trappc4, but significantly reduced overall and membrane-bound levels of trappc9, which specifies TRAPP toward activating Rab11. Trappc9 deficiency led to elevated expression of kalirin in neurons. Co-localization of kalirin and Rab11 occurred at a low frequency in NRK cells under steady state and was enhanced upon expressing an inactive Rab11 mutant to prohibit the dissociation of Rab11 from the kalirin-TRAPP complex. The small RNA-mediated depletion of kalirin diminished activities in cellular membranes for activating Rab11 and resulted in a shift in size of Rab11 positive structures from small to larger ones and tubulation of recycling endosomes. Our study suggests that kalirin and TRAPP form a dual GEF complex to choreograph actions of Rab11 and Rac1 at recycling endosomes.

## 1. Introduction

Many critical cellular processes require a tight integration of vesicle-mediated membrane dynamics with the rearrangement of local actin cytoskeleton networks. For example, cell division and polarized cell migration, key events of cancer propagation and metastasis, demand vesicular supply of necessary membrane molecules and local actin cytoskeleton reorganization to adjust the cell shape [1]. In the nervous system, the induction of long-term potentiation, a crucial mechanism of learning and memory, involves activity-dependent growth of dendritic spines, which relies on vesicle mediated addition of extra membranes containing neurotransmitter receptors (functional plasticity) and rearrangement of actin filaments within the spines (structural plasticity) to retain the synapses [2,3]. The budding of transport vesicles also involves the integration of a Rab-engaged assembly of vesicle machinery and Rho-orchestrated polymerization of actin networks at budding sites of donor membranes [4]. How the rearrangement of local actin cytoskeleton networks is promptly integrated with dynamic changes of membranes remains largely unknown. 

The Rab family of GTPases participate in the four major steps of vesicular trafficking: vesicle formation, delivery, tethering, and fusion [5]. The Rho family of GTPases including Rac1 and cdc42 play pivotal roles in remodeling actin filaments to shape and move cells as well as intracellular organelles and vesicles [6,7]. Therefore, an attractive mechanism for the timely remodeling of local actin cytoskeletal networks in concert with dynamic changes of membranes may be efficient coordination of Rho and Rab GTPases. Indeed, evidence is emerging that reveals an interplay between Rab and Rho GTPases [8,9].

Rab and Rho families of GTPases are molecular switches that alternate between GDP-bound inactive and GTP-bound active states and exert functions through a distinct set of downstream effector proteins [5,6,7]. The balance between the two states of a Rab/Rho GTPase is under tight control of a guanine nucleotide exchange factor (GEF) and a GTPase-activating protein (GAP), which switch the cognate Rab/Rho on and off, respectively. Previous studies suggest that the interplay between Rab and Rho GTPases may occur at different levels. For instance, Rac1 regulates Rab5 and Rab7 by interacting with Rab5GEF alsin and Rab7GAP Armus, respectively, whereas Rab6 modulates cdc42 by interacting with RhoGEF trio [10,11,12]. 

Unlike the aforementioned mechanisms involving sequential actions of Rab and Rho, we report herein a dual GEF complex for concerted actions of Rab and Rho. This GEF complex comprised of kalirin, a member of the diffuse B-cell lymphoma homologue (DH) domain family of RhoGEFs activating Rac1 [13], and transport protein particle (TRAPP), a modular multiunit complex acting as a GEF for Rab proteins [14,15]. Kalirin was identified from proteins precipitated by trappc4, one of the four subunits forming a core that confers the GEF activity onto TRAPP [16], and located on trappc4 containing cellular organelles. Depletion of kalirin downregulated the overall expression and membrane association of trappc9, which specifies TRAPP toward activating Rab11, diminished Rab11GEF activities on cellular membranes, and induced signs of impaired production of transport vesicles from recycling endosomes. Our study suggests that TRAPP and kalirin form a complex to facilitate the actions of Rab11 and Rac1 at recycling endosomes.

## 2. Materials and Methods 

### 2.1. Plasmid DNA

The construction of the plasmids expressing GST-Rab1-6xHis, GST-Rab11-6xHis, GST fused Rab11 and its mutants were described previously [17]. The cDNAs encoding for mouse trappc4 were amplified by reverse transcriptase polymerase chain reaction and cloned into pGEX-6p, from which GST-fused trappc4 or GST alone was sub-cloned into pcDNA_3_. The plasmids expressing Kalrn23-684, Kalrn674-1272, and Kalrn1269-1654 were generated as previously described [18]. All constructs were verified by DNA sequencing. 

### 2.2. Expression and Purification of Tagged Proteins

GST-Rab11, GST-dNRab11, GST-dARab11, GST-Rac1, and GST-trappc4 were expressed in bacteria and affinity purified using glutathione resins (GE Healthcare Life Sciences) according to manufacturer’s instructions as described previously [17,19,20]. GST-Rab1-6xHis and GST-Rab11-6xHis proteins were purified by sequentially incubation with glutathione and Ni-NTA resins (QIAGEN). All purified proteins were dialyzed against 20 mM HEPES-Na, pH 7.4, 100 mM KCl, 1 mM DTT and 2 mM EDTA to remove nucleotides from Rabs and stored in aliquots at −80 °C until use. 

### 2.3. Cell Culture and Transfection

All media and supplements were purchased from Gibco (ThermoFisher), whereas fetal bovine serum was obtained from Atlanta Biologicals. NRK and HEK293T cells were maintained in DMEM supplemented with 10% fetal calf serum, L-glutamine and penicillin/streptomycin at 37 °C in a humidified incubator (ThermoFisher). The day before transfection, cells were plated on coverslips in a 6-well plate or in a 60mm plate at a concentration of 2 × 10^5^ ml^-1^. Prior to the preparation of liposome-DNA/siRNA mixtures, cells were changed into Opti-MEM medium (ThermoFisher) and cultured for 2 to 3 hrs. For RNA interference, siRNAs for kalirin (acaggaagacctacggaaa, DHARMACON) or eGFP (DHARMACON) were mixed with empty pcDNA_3_ at a ratio of 27 ng of siRNAs and 1 µg of pcDNA_3_ in 100 µl of Opti-MEM and combined with 5 (per well of a 6-well plate) or 10 (per 60mm plate) µl of lipofectamine^2000^ diluted in 100 µl of Opti-MEM. The liposome-siRNA/pcDNA_3_ mixtures were incubated at room temperature for 25 min with gently mixing every 5 min and evenly applied to cells. Cells were cultured in total of 0.5 ml (per well of a 6-well plate) or 1.5 ml (per 60mm plate) of Opti-MEM media for 5 hrs and then changed into complete media. Cells were cultured for additional 48 hrs and then used for analysis. For introducing dNrab11 into NRK cells, 0.3 µg of pcDNA_3_-Rab11S25N were used for transfection. After adding the liposome-DNA mixtures, cells were cultured in an incubator for 5 hrs and then processed for immunofluorescence labeling. 

For discovering proteins potentially associated with mTRAPP, HEK293T cells were transfected with pcDNA_3_ expressing GST-trappc4 or GST alone for 48 hrs. Total cellular membranes were prepared and used for precipitating protein complexes containing GST-trappc4 or GST alone. Proteins bound onto glutathione resins were analyzed by SDS-PAGE followed by silver staining or Western blot. Gel slices containing silver stained proteins were subjected to mass spectrometry at the Harvard Medical School Taplin Mass Spectrometry Core Facility. For determining the domain(s) at which trappc4 was associated with kalirin, 293T cells were used for transfection with plasmids expressing different fragments of kalirin. Post-nuclear supernatants from cells transfected with each of the plasmids expressing kalirin domains were incubated with bacterially expressed and affinity purified GST-trappc4. Proteins bound to GST-trappc4 were captured on glutathione resins, washed, and eluted into appropriate buffers for analysis. 

### 2.4. Preparation of Total Cellular Membranes

Lysates of cultured cells were prepared by passing cell suspensions in homogenate buffer containing 25 mM Tris/Cl, pH 7.4, 0.25 M sucrose, and protease inhibitors through 271/2-gauge needles for 15 strokes. Homogenates were centrifuged at 12,000 rpm for 5 min and the pellet (P1) was discarded. The resulting post-nuclear supernatant (S1) was centrifuged at 55,000 rpm for 1 hr in a TLA100.2 rotor (Beckman Coulter). The pellets (P2) were designated as total cellular membranes and solubilized in lysis (25 mM Tris/Cl, pH 7.4, 150 mM NaCl, 1 mM EDTA, and 1 mM DTT) or GEF buffers containing 0.5% of triton X-100 and protease inhibitors. Insolubilized membranes were removed by a centrifugation at 5000 rpm for 5 min. After determining the protein concentrations, solubilized membranes were used for pull-down assays or GEF reactions.

### 2.5. Guanine Nucleotide Exchange Assay

[^3^H]GDP release and subsequent quantification were performed as previously described [17,19,20]. In brief, 0.5 μg of GST and 6 x His fused Rab proteins (Rac1, Rab1 and Rab11) were loaded with 20 picomol of [^3^H]GDP (11.9 Ci/mmol, Amersham) in loading buffer (20 mM HEPES, pH 7.2, 20 mM potassium acetate, 1 mM DTT, 5 mM EDTA) at 37 °C for 30 min and then immobilized on Ni-NTA beads after adding MgCl_2_ to a final concentration of 10 mM. Free [^3^H]GDP was removed by washing Ni-NTA beads twice in ice-cold loading buffer containing 10 mM MgCl_2_. GEF samples (immunoprecipitates or cellular membranes) were diluted or solubilized in 50 μl of assay buffer containing 20 mM HEPES, pH 7.2, 5 mM Mg(OAc)_2_, 1 mM DTT, and 0.75 mM GTP/GDP and added to corresponding [^3^H]GDP-loaded Rab proteins. The nucleotide exchange reaction was initiated by incubating in a 37 °C water bath with gentle mixing for 30 min and stopped by transferring the reaction mixtures on ice. Resins with Rabs were collected, washed twice in cold buffer (20 mM Tris/Cl, pH 7.4, 20 mM NaCl, 5 mM MgCl_2_ and 1 mM DTT), and then transferred into scintillation vials for counting. Data were represented as mean percentage of [^3^H]GDP remaining on Rab proteins.

### 2.6. Transferrin Recycling

NRK cells transfected with siRNAs were changed into serum-free medium and cultured at 37 °C for 1 hr to maximally deplete the remaining serum. Cells were washed three times in cold PBS, incubated in ice-cold serum-free medium on ice for at least 30 min to completely cool down cells, and then changed into fresh ice-cold serum-free medium containing 5 µg ml^-1^ Alexa568-transferrin (Invitrogen). After an incubation on ice for 30 min and two washes in cold PBS, cells on coverslips were transferred into pre-warmed complete medium containing 10 µg ml^-1^ unlabeled holo-transferrin and incubated at 37 °C for indicated times. After transferrin uptake, cells were washed in cold PBS, fixed in cold (−20 °C) methanol for 1 min, and processed for confocal microscopic analysis. Digital images were collected using the “peak” mode with the same settings for each of the incubation and transfection conditions and opened in NIH ImageJ for measuring signal intensities of Alexa568-transferrin. Mean intensities of background signals for each image were measured and removed from intensities of Alexa568-transferrin signals. Mean ± SD intensities of Alexa568-transferrin per cell for each condition were calculated for comparison.

### 2.7. Immunohistochemistry and Western Blot

Immunofluorescence analysis was performed as standard procedures. In brief, cells on glass coverslips were washed twice in PBS, fixed in 4% paraformaldehyde/PBS at room temperature for 15 min, quenched in 50 mM NH_4_Cl, washed twice in PBS and labeled with primary antibodies against *pan*-kalirin (polyclonal against the 4 to 7 spectrin repeats of kalirin; Millipore) and Rab11 (monoclonal against AA86-207 of human Rab11a; BD Biosciences). Targeted proteins were detected by incubating cells on glass coverslips with secondary antibodies conjugated with BODIPY or Cy-3 (Jackson Laboratory). Microscopy was performed using 60 x or 100 x oil Nikon Plan Apo objective mounted on an inverted Nikon Eclipse TE300 fluorescent microscope. Images of each channel were collected separately using the “peak signal” mode through a Bio-Rad Laser-sharp confocal system equipped with krypton-argon and blue diode lasers, and merged using PhotoShop. Images for the comparison of signal intensities and or endosome size were collected with the same settings including laser strength, pinhole, signal gain, resolution, and scan times. For quantitative analyses, intensities of fluorescent signals and/or cross-sectional areas of each imaged cell were measured with the NIH ImageJ.

For measuring the size of Rab11 positive endosomes, digital images were opened with NIH ImageJ. After appropriately adjusting the signal threshold, we applied the “Analyze Particles” tool to identify Rab11 positive endosomes and manually measured the identified “abnormally large particles”. The value of the measured area of each “particle” was converted to nm^2^ according to the original size and resolution of the image. We grouped Rab11 endosomes (particles) based on their size in nm^2^ and calculated their percentages for graphing and comparison. Those Rab11 positive structures with a length more than 3 folds of the width were considered as tubulated endosomes. A Chi square test for trend was conducted to determine statistical significance between eGFP-siRNA and kalirin-siRNA treatment conditions. 

SDS-PAGE and Western blot analysis were performed as standard procedures. Concentrations of primary antisera were used as following: polyclonal anti-trappc9 (1:1000, Abcam, ab104041), polyclonal antibodies for *pan*-kalirin (1:750), monoclonal anti-trappc4 (1:500, Santa Cruz, C-7), monoclonal anti-HA (1:2000, Roche), monoclonal anti-GAPDH (1:1000, ProteinTech), and monoclonal anti-actin (1:1000, Sigma). Peroxidase-conjugated secondary antibodies (Jackson Laboratories) were diluted at 1:5000. The blots were developed using enhanced ECL (Pierce). Films were scanned and densitometry data were obtained using the NIH ImageJ.

### 2.8. Cryo-Immunogold Electron Microscopy

NRK cells were fixed for 1 hr at room temperature by directly adding freshly prepared 4% formaldehyde and 0.2% glutaraldehyde (Electron Microscopy Sciences, Hatfield, PA) in 0.1 M phosphate buffer (pH 7.4) to cells in a 10cm dish immediately after discarding cell culture medium. After 4 washes in PBS, cells were gently scraped into PBS and pelleted by a centrifugation at 1200 rpm for 10 min. The cell pellet was solidified in 2% agarose in PBS on ice and cut into small blocks. Blocks were cryo-protected by an overnight infiltration with 2.3 M sucrose in PBS at 4 °C and then mounted on aluminum pins and frozen in liquid nitrogen. Ultrathin cryosections were cut on a Leica EM FCS at –80 °C, collected on formvar-coated nickel grids, and floated on PBS with 0.02% sodium azide as a preservative. Double immunogold labeling with the protein A-gold method was done as described previously [21]. In brief, grids were blocked with 1% BSA in PBS and incubated with primary antibody for 1 hr at room temperature followed by incubation with protein-A conjugated with 5 nm gold particles for 1 hr. After rinses with PBS, the grids were fixed for 5 min on a drop of 1% glutaraldehyde in PBA, rinsed, blocked with 1% BSA in PBS, and labeled with the second primary antibody followed by incubation with protein-A conjugated with 15 nm gold particles. After several rinses on drops of distilled water, the grids were floated on drops of Tylose® cellulose and uranyl acetate for 10 min on ice, collected on loops and allowed to dry. The grids were examined at 80 kV in a JEOL JEM 1011 transmission electron microscope. Images were acquired using an AMT digital imaging system (Advanced Microscopy Techniques, Danvers, MA).

## 3. Results

### 3.1. Kalirin Interacts with mTRAPP

TRAPP is a modular complex identified initially as a vesicle tether in yeast and acts as a GEF for Ypt (Rab) GTPases [22,23]. Mammals express all homologs of yeast (y) TRAPP subunits, in addition to specific subunits that do not exist in fungal cells [15]. There are two mammalian (m) TRAPP complexes termed mTRAPP-II and mTRAPP-III, which are specified by trappc9/10 and trappc8/11/12/13, respectively, and play opposing roles in regulating intracellular ricin transport [24]. However, the function of mTRAPP remains largely unknown. To look for proteins potentially modulating the function of mTRAPP, we performed pulldown experiments using glutathione S-transferase (GST) tagged trappc4, one of the four subunits forming a core that confers GEF activities onto TRAPP. SDS-PAGE followed by silver staining detected several protein bands that were present in precipitates obtained with GST-trappc4 but not with GST alone (Figure 1A). Mass spectrometry analysis of gel slices containing the silver stained proteins precipitated by GST-trappc4 led to the identification of mTRAPP subunits trappc9 and trappc10 as well as kalirin (upper panel, Figure 1A), which is a typical member of the DH family of Rho GEFs and activates Rac1 [25]. The *kalirin* gene in human encodes 2986 amino acids. Alternative splicing generates different kalirin isoforms, including a 190kD neuronal isoform (kalirin-7) and a 217kD non-neuronal isoform (kalirin-8) [25].

The finding of RhoGEF kalirin co-precipitated with RabGEF mTRAPP subunit trappc4 prompted us to hypothesize that kalirin and mTRAPP might form a complex to facilitate actions of Rho and Rab GTPases. To test this hypothesis, we first verified the association of kalirin with trappc4 by Western blot analysis with antibodies specific for kalirin (lower panel, Figure 1A) and further determined the region in kalirin that is required for the association with trappc4 using constructs expressing kalirin domains (Figure 1B). Our pulldown studies showed that Kalrn23-684 was co-precipitated with GST-trappc4 (Figure 1C). Kalrn1269-1654, which was expressed at levels similar to Kalrn23-684, could not be precipitated by GST-trappc4 (Figure 1C). Longer exposures of the blots revealed that Kalrn674-1272 was also pulled down by GST-trappc4 (data not shown). As huntingtin interacts with both Kalrn674-1272 and a Rab11GEF [18,19], the co-precipitation of Kalrn674-1272 with GST-trappc4 is likely because huntingtin was present in the precipitates. These results demonstrate the bona fide association between kalirin and mTRAPP. We then carried out electron microscopic studies to expose whether kalirin and trappc4 (mTRAPP) exerted functions on the same organelles. Double immunogold labeling of ultrathin sections of NRK cells showed that kalirin and trappc4 (mTRAPP) were co-localized at tubulovesiclular membranes (Figure 1D). These data support that kalirin and mTRAPP act together and form a complex in cells.

### 3.2. Kalirin is Associated with mTRAPP-II

Having shown the association of kalirin with mTRAPP, we then examined if kalirin and mTRAPP affected each other. We first determined if kalirin modulated the expression and/or membrane association of mTRAPP subunits. We transfected NRK cells with small interfering RNAs (siRNA) specific for kalirin or enhanced green fluorescent protein (eGFP). Western blot analysis showed that expression levels of kalirin were significantly reduced in cells transfected with kalirin-specific siRNAs, relative to those in cells treated with eGFP-specific siRNAs (Figure 2A,B). Acute knockdown of kalirin did not influence the expression or membrane association of trappc4, a subunit shared both mammalian TRAPP complexes, but significantly diminished overall as well as membrane-bound levels of mTRAPP-II specific trappc9 (Figure 2A,B). To determine if trappc9 affected the expression of kalirin, we examined brain lysates of trappc9 null mice that we generated recently. Western blot analysis followed by densitometry quantification revealed that expression levels of the neuronal isoform of kalirin (190kD) were significantly elevated in the brain of trappc9 null mice compared to WT mice (Figure 2C,D). Collectively, these data support that kalirin is associated with mTRAPP-II.

### 3.3. Kalirin Regulates the Function of Rab11

As knockdown of kalirin downregulates trappc9, which specifies TRAPP toward activating Rab11 [27,28], we reasoned that kalirin might participate in activating Rab11. In support of this idea, immunoprecipitates obtained with antibodies specific for overexpressed kalirin contained GEF activities on Rab11 but not Rab1 (Figure 3A). In addition, cellular membranes from cells transfected with siRNAs for kalirin contained significantly less support for releasing [^3^H]GDP from GST-Rab11-6xHis but not GST-Rab1-6xHis than the same amount of cellular membranes from cells transfected with eGFP-specific siRNAs (Figure 3B). Together with the finding that kalirin itself has no GEF activity onto Rab11 [18], we suggest that kalirin regulates Rab11 through association with mTRAPP-II. We then examined if knockdown of kalirin affected transferrin (Tfn) recycling, a cellular process widely used for determining the function of Rab11 [29]. We performed synchronized Alexa568-Tfn uptake and recycling as previously described [30]. When synchronized Tfn uptake was conducted for 5 min, there was no difference in the signal intensities of intracellular Alexa568-Tfn between cells treated with kalirin-siRNA and cells transfected with eGFP siRNA (Figure 3C). When the uptake time was prolonged to 30 min, signals of Alexa568-Tfn markedly declined in GFP-siRNA treated cells, but still accumulated in cells transfected with siRNAs for kalirin (Figure 3C). These data suggest that kalirin depletion does not influence endocytosis, but impairs endosomal recycling. Taken together, these results support that kalirin regulates the function of Rab11.

### 3.4. Kalirin Functions at Rab11 Positive Recycling Endosomes

To further demonstrate that kalirin regulates Rab11 under physiological conditions, we examined if kalirin located at Rab11 positive endosomes. Double immunofluorescence labeling revealed that kalirin and Rab11 were co-localized at punctate structures in NRK cells, but the frequency of their co-localization at steady state was low (Figure 4A). We reasoned that the low frequency of the co-localization resulted from the rapid dissociation of Rab11 from the kalirin-mTRAPP-II complex upon being activated by the GEF complex, but was not a consequence of the high density of structures in cells. If this speculation is true, approaches to prohibit the dissociation of Rab11 from the GEF complex should significantly increase the frequency of the co-localization between kalirin and Rab11. Since dominant negative Rab mutants are high-affinity ligands of their GEFs, we introduced Rab11S25N, a dominant negative Rab11 mutant, to hinder the dissociation of Rab11 from the GEF complex. As expected, expression of Rab11S25N at low levels vastly increased the frequency of the co-localization of kalirin and Rab11 (Pearson’s correlation co-efficient: 0.748 ± 0.06, n = 13 cells, Figure 4B). To rule out the possible contribution of secondary antibodies to the “rare” co-localization events, we switched red/green dye-conjugated secondary antibodies for detecting Rab11 and kalirin and obtained similar results (cell indicated by a star symbol, Figure 4B). These data support that kalirin does locate at Rab11 positive recycling endosomes.

### 3.5. Knockdown of Kalirin Causes Tubulation of Recycling Endosomes

Having shown that kalirin locates at recycling endosomes and regulates Rab11 and Tfn receptor recycling, we next explored which step(s) of endocytic recycling is likely to involve kalirin. As Rab11 is suggested to control vesicle formation at recycling endosomes [31], we wanted to determine if kalirin took part in vesicle formation at recycling endosomes. If so, the depletion of kalirin should cause morphological changes of Rab11 positive recycling endosomes similar to those induced by Rab11 dysfunction. Treatment with kalirin specific siRNAs led to a vast decrease of signals immunoreactive for kalirin and a change of the distribution of kalirin signals from a diffuse pattern, as in cells treated with eGFP specific siRNAs, to a pattern of being concentrated in large-sized punctate structures, which also contained Rab11 (Figure 5A). Consistent with previous findings of spine shrinkage when the expression or function of kalirin is reduced [32], the size of cells treated with kalirin specific siRNAs was significantly reduced relative to that of cells treated with siRNAs for eGFP (Figure 5A,B). Long tubular structures positive for Rab11 were barely seen in cells transfected with eGFP-specific sRNAs, but frequently observed in clusters in cells transfected with siRNAs for kalirin (Figure 5A). Further analysis revealed that there was a shift in the size of Rab11 positive structures from small to larger ones in cells transfected with kalirin siRNAs relative to those in cells transfected with eGFP siRNAs (Figure 5C). These data suggest that knockdown of kalirin is likely to induce a deficit in the production of transport intermediates from recycling endosomes.

## 4. Discussion

In this study, we report a dual GEF complex formed by RhoGEF kalirin and RabGEF mTRAPP for coordinating Rab and Rho GTPases. Kalirin is co-precipitated and co-localized with trappc4, a subunit present in both of the two mammalian TRAPP complexes, namely mTRAPP-II and mTRAPP-III. However, kalirin depletion does not influence trappc4, but downregulates trappc9, which specifies mTRAPP-II for activating Rab11, and in accordance reduces Rab11GEF activities on cellular membranes. Immunoprecipitates obtained with antibodies for kalirin contain GEF activities onto Rab11, but not Rab1, which is activated by mTRAPP-III. These lines of evidence suggest that kalirin interacts with mTRAPP-II and regulates Rab11. In agreement with this idea, co-localization between kalirin and Rab11 occurs at steady state and is boosted upon expressing a dominant negative Rab11 mutant. Additionally, abrogation of trappc9 in mice elevates the expression of kalirin in neurons in the brain. As trappc9 deficiency impairs the formation of mTRAPP-II with no effect on mTRAPP-III (data not shown). The upregulated expression of kalirin may be a compensatory effect developed for stabilizing the fragmental “mTRAPP-II” in trappc9-void cells. Together with the well-known role of kalirin in activating Rac1, we conclude that kalirin and mTRAPP-II form a complex to coordinate actions of Rac1 and Rab11 at recycling endosomes. 

Interplay between Rab11 and Rac1 has been demonstrated in a previous study [33]. In this case, Rab11 through its effector FIP3 recruits Rac1 onto recycling endosomes and regulates the clustering of Rac1 at immunological synapses in T cells. It remains unclear whether the recruitment of Rac1 onto recycling endosomes is adopted as a way to keep Rac1 away from the plasma membrane [9,33], or required for normal functions of recycling endosomes. Actin polymerization induced by the Arp2/3 complex activator WASH controls the fission of transport intermediates or vesicles from recycling endosomes [34]. However, the regulators acting upstream of WASH are not known. Our studies expose evidence supporting that similar to WASH depletion, kalirin depletion induces defective transferrin recycling and tubulation of recycling endosomes. Collectively, we suggest that kalirin and Rac1 are the “missing” regulators functioning upstream of WASH at Rab11 positive recycling endosomes.

Kalirin and mTRAPP-II are not loosely gathered together for sequentially activating their respective substrates, Rac1 and Rab11, specifically, but form an interdependent complex, as implicated by the findings of kalirin and trappc9 mutually affecting their expression levels. The kalirin-mTRAPP-II complex may be highly dynamic and frequently undergo disassembly-reassembly cycles in a temporal and spatial manner. Other protein(s) may join in and modulate the dynamics and function of the kalirin-mTRAPP-II complex. For an example, the Huntington’s disease causative protein huntingin, which interacts with kalirin at a middle domain (Kalrn674-1272) [18], is a good candidate. Previous studies have shown that both the loss of huntingtin and the presence of the disease mutation in huntingtin compromise a GEF in activating Rab11 [17,19,20]. However, the identity of the compromised GEF has not been unveiled. In view of huntingtin and mTRAPP-II binding to different domains in kalirin, we suggest that mTRAPP-II is the GEF compromised in Huntington’s disease. In this instance, our study provides a key clue for future studies to devise strategies to maneuver Rab11 dysfunction in Huntington’s disease. Huntingtin is well known to interact with various proteins and may potentially recruit these binding partners to the kalirin-mTRAPP-II complex to further diversify the cellular processes involving this dual GEF complex.

In addition to our work, there is support from other studies that kalirin functions with Rab11 in endocytic recycling. Disturbance of kalirin activity causes a deficit in the delivery of α-amino-3-hydroxy-5-methyl-4-isoxazolepropionic acid (AMPA) receptor from intracellular compartments to synaptic surfaces following long-term potentiation (LTP) [35,36,37,38]. Knockdown of kalirin in neurons results in accumulation of AMPA receptors at the base of spines [38]. Recycling endosomes supply AMPA receptors for LTP [35,36,37] and are positioned at the base of spines [37]. Rab11 activity is critical for the delivery of AMPA receptors from recycling endosomes to synaptic surfaces [35,37]. Furthermore, expression of a dominant negative Rab11 mutant to interfere with the function of Rab11 in neurons is sufficient to render spine loss and perturbation of activity-evoked spine enlargement [37]. These common regulatory properties of kalirin and Rab11 in neurons support our idea that kalirin regulates the function of Rab11. 

In summary, our biochemical and cell biological studies establish that Rac1GEF kalirin forms a dual GEF complex with Rab11GEF mTRAPP-II to regulate the function of Rab11 and the fission of recycling endosomes. This dual GEF complex is likely to be compromised and responsible for impaired functions of recycling endosomes in Huntington’s disease, and thus provides a target for future studies to devise intervention.

## Figures and Tables

**Figure 1 cells-09-01132-f001:**
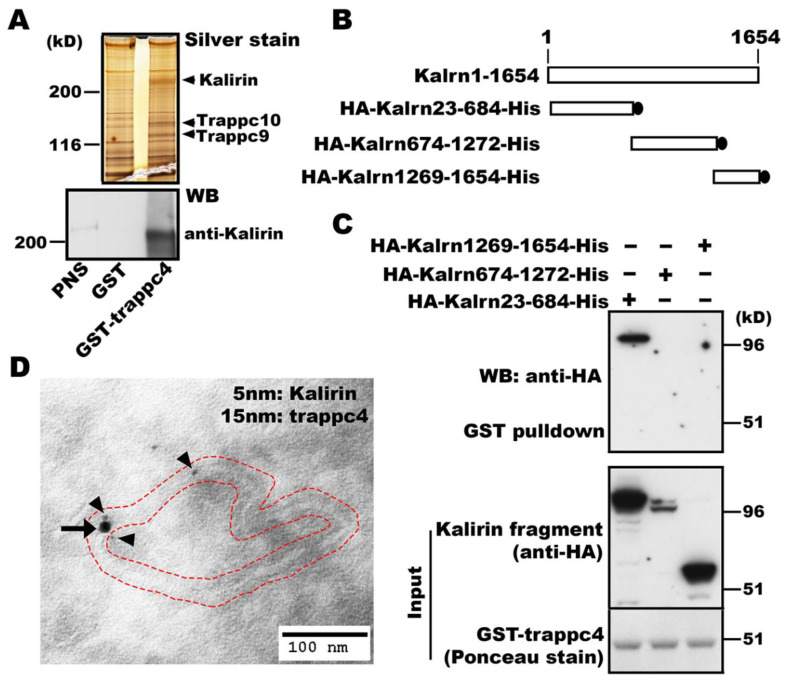
Kalirin is associated with trappc4. (**A**) Mass spectrometry analysis of proteins associated with GST-trappc4. Solubilized cellular membranes from 293T cells transfected with pcDNA_3_ expressing GST-trappc4 or GST alone were incubated with glutathione resins. Proteins on resins were analyzed by SDS-PAGE followed by silver staining (upper panel) or Western blot (lower panel). Upper panel: Arrows indicate proteins identified from gel slices precipitated by GST-trappc4 but not by GST alone. Lower panel: Western blot analysis with antibodies specific for kalirin verifies the association of kalirin with GST-trappc4 but not with GST alone. (**B**) Schematic representation of constructs expressing different domains of kalirin. The amino acid sequence for generating the constructs was based on rat kalirin-7 [26]. The number of amino acids is shown above. (**C**) Association of the N-terminal portion (Kalrn23-684) of kalirin with trappc4. Post-nuclear supernatants of 293T cells transfected with plasmids expressing the indicated regions of kalirin were incubated with GST-trappc4 pre-immobilized on glutathione resins. After washes, proteins on resins were analyzed by Western blot with indicated antibodies. Shown are data from one of three experiments with similar results. (**D**) Co-localization of trappc4 with kalirin at tubulovesicular membranes. Double immunogold labeling of ultrathin sections of NRK cells was performed as in Methods. Arrowheads point to labeling of endogenous kalirin (5 nm gold particles), whereas the arrow indicates labeling of endogenous trappc4 (15 nm gold particle) in proximity to kalirin labeling at a tubulovesicular membrane structure, the membranes of which lie in between the two dashed red tracing contours.

**Figure 2 cells-09-01132-f002:**
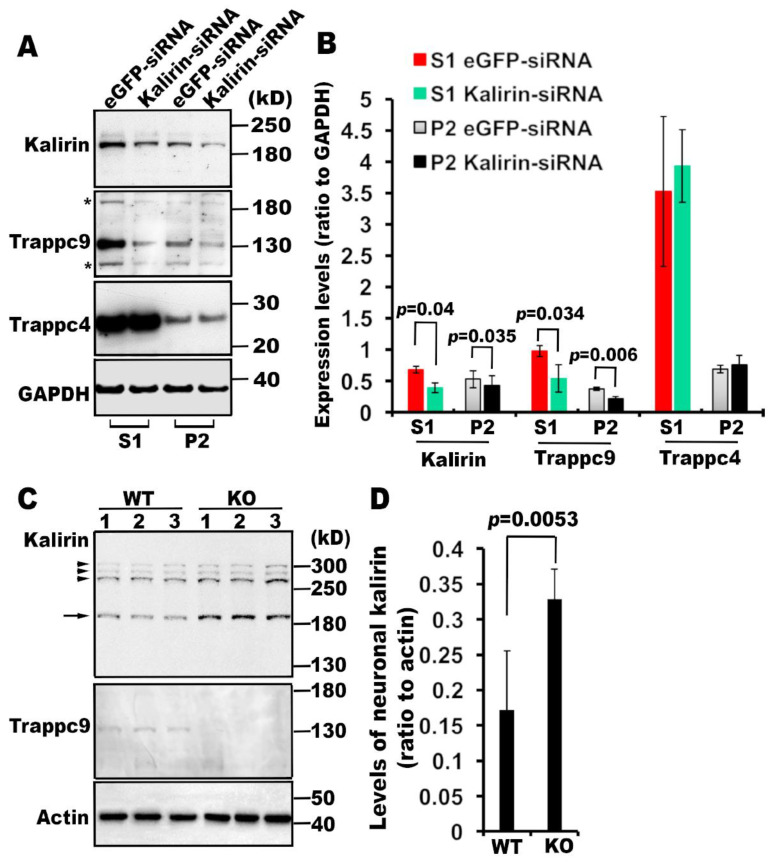
Kalirin and trappc9 mutually affect their expression. (A) and (B) Kalirin knockdown perturbs the expression as well as membrane association of trappc9. (**A**) Post-nuclear supernatants (S1) and cellular membranes (P2) from NRK cells treated with siRNAs for eGFP or kalirin were analyzed by Western blot with indicated antibodies. Shown are blot analyses from one of three experiments with similar results. Note that the blots were re-probed with anti-trappc9 antibodies after incubation with anti-kalirin antibodies, therefore signals detected by anti-kalirin antibodies still persisted as indicated by stars (*). (**B**) Films with signals in linear ranges were scanned into digital images for measuring intensities of signals for each protein and corresponding background signals using NIH ImageJ. Data are expressed as ratio of signal intensity of the indicated protein to that of GAPDH (n = 3, Mean ± SD, two-tailed Student’s *t*-test). (**C**) and (**D**) Constitutive loss of trappc9 causes elevated expression of kalirin in brain neurons. (**C**) Western blot analysis of post-nuclear supernatants from brain tissues of three wild-type (WT) and three trappc9 knockout (KO) mice. The genotype of all mice was determined by PCR. The age of all mice used for harvesting brain tissues was 3 months. The arrow points to the neuronal isoform of kalirin (190 kD), whereas arrowheads indicate other kalirin isoforms. (**D**) Intensities of signals for the neuronal kalirin isoform as well as actin and background in the same lane were measured with NIH ImageJ as above and used for calculating Mean ± SD ratio of neuronal kalirin to actin signals. Comparison was performed with two-tailed Student’s *t*-test.

**Figure 3 cells-09-01132-f003:**
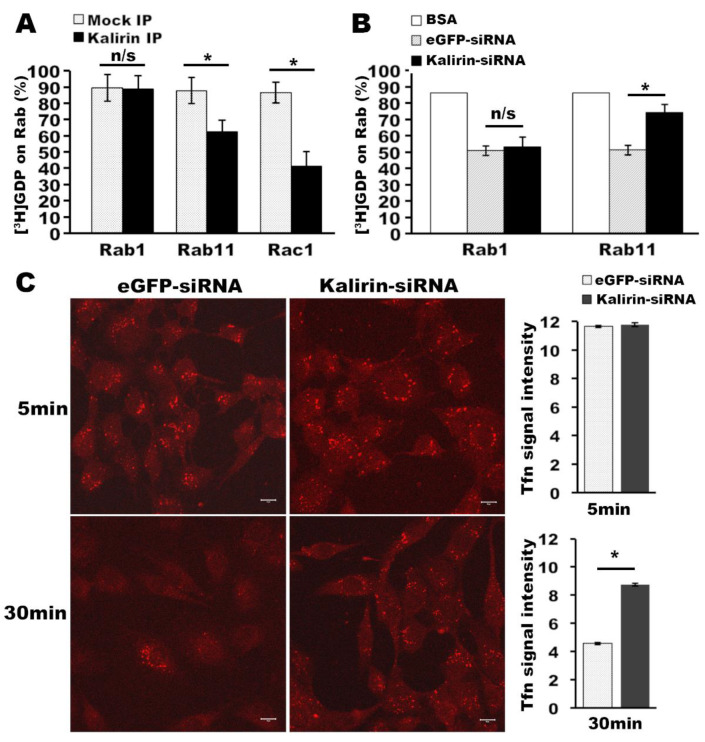
Kalirin regulates the activation of Rab11 and endocytic recycling of transferrin. (**A**) Kalirin immunoprecipitates containing GEF activities on Rab11. Triton solubilized cellular membranes from cells transfected with pEAK-His-myc-kalirin7 or empty vector were used for immunoprecipitation with anti-myc antibodies. Precipitated proteins were used for the [^3^H]GDP release assay as described in Methods. Mean ± SD percentages of [^3^H]GDP remaining on GST-Rab1-His, GST-Rab11-His, and GST-Rac1 were graphed (n = 3, two-tailed Student’s *t*-test: n/s, not significant, * *p* < 0.01). (**B**) Knockdown of kalirin reduced Rab11GEF activities in cellular membranes. After knocking down kalirin in NRK cells as in (2A), cellular membranes were prepared and extracted in GEF assay buffer containing triton X-100. Equal amounts of solubilized membranes from cells treated with eGFP-siRNA and kalirin-siRNA were used for [^3^H]GDP release from GST-Rab1-His and GST-Rab11-His as above. BSA is a no-GEF control (n = 3, Mean ± SD, two-tailed Student’s *t*-test: n/s, not significant, * *p* < 0.01). (**C**) Synchronized uptake and recycling of Alexa568-transferrin was performed with siRNA-treated NRK cells for indicated times. After uptake, NRK cells on glass coverslips were processed for fluorescent microscopy. Shown are confocal images. Scale bar: 10 μm. Signal intensities of Alexa568-transferrin and the corresponding background of the images for each condition were measured using NIH ImageJ. After subtracting the background, the mean values of Alexa568-transferrin signals were calculated and plotted (n = 33 cells for kalirin-siRNA and 34 cells for eGFP-siRNA for 5 min incubation, n = 35 cells for kalirin-siRNA and 31 cells for eGFP-siRNA for 30min incubation; Mean±SD, two-tailed Student’s *t*-test: * *p* < 0.01).

**Figure 4 cells-09-01132-f004:**
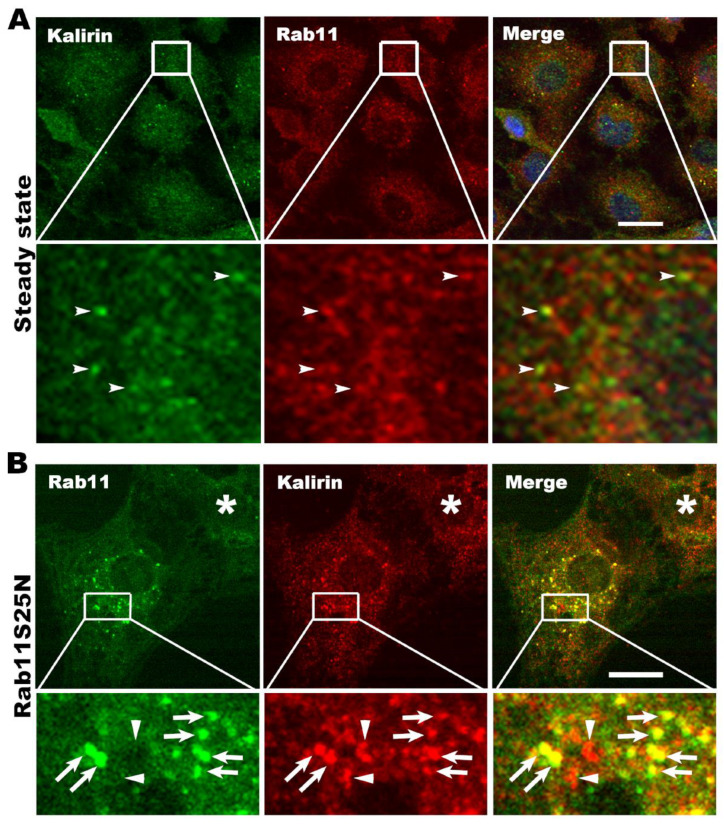
Kalirin locates at Rab11 positive recycling endosomes. (**A**) Low frequency of co-localization between kalirin and Rab11 at steady state. NRK cells on glass coverslips were fixed and processed for labeling with antibodies for kalirin (green) and Rab11 (red). Shown are confocal images. Boxed regions are enlarged and shown underneath the corresponding image/channel. Arrowheads identify structures co-labeled with Rab11 and kalirin. Scale bar: 10 μm. (**B**) Enhanced co-localization of kalirin with Rab11 upon expression of Rab11S25N, a dominant negative mutant of Rab11. NRK cells on glass coverslips seeded in a 6-well plate were transfected with 0.3 μg of pcDNA_3_-Rab11S25N. After 5 hrs of culture, cells were processed for labeling Rab11 (green) and kalirin (red). Boxed regions are enlarged and shown underneath the corresponding image/channel. Arrows indicate structures containing both Rab11 and kalirin, whereas arrowheads identify Rab11 positive structures void of kalirin. The star symbol in each channel indicates a cell likely to be not transfected. Scale bar: 5 μm.

**Figure 5 cells-09-01132-f005:**
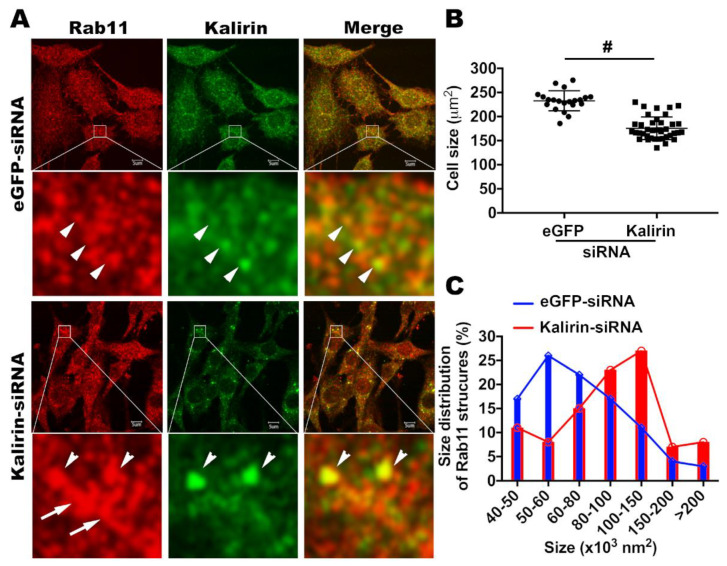
Knockdown of kalirin induces cell shrinkage and tubulation of recycling endosomes. NRK cells were transfected with siRNAs for eGFP or kalirin as above and processed for labeling Rab11 (red), kalirin (green), and nuclei (blue). (**A**) Confocal images show that signals for kalirin were diffusely distributed and occasionally occurred at Rab11 positive structures in NRK cells treated with eGFP-specific siRNAs, whereas kalirin signals were concentrated at large punctate structures co-labeled with Rab11 in kalirin-siRNA treated NRK cells. Boxed areas are enlarged and shown beneath the corresponding images. Arrowheads point to structures containing both Rab11 and kalirin. Arrows in the red channel of kalirin-siRNA treated NRK cells identify elongated Rab11 positive recycling endosomes, which are rarely seen in eGFP-siRNA treated cells. Scale bars: 5 μm. (**B**) Comparison of cross-sectional areas of cells treated with eGFP-siRNA *vs* kalirin-siRNA. Each symbol in the plot represents one cell (Mean ± SD, Student’s *t*-test: #: *p* < 0.0001). (**C**) Comparison of Rab11 positive structures shows that there is a shift in size of Rab11 positive structures from smaller to larger ones in kalirin-siRNA treated cells relative to those in cells treated with eGFP-siRNA. Statistical significance was conducted with Chi square test for trend (*X*^2^, df = 16.99, 1: *p* < 0.0001).

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
