# Peer review of "Kalirin Interacts with TRAPP and Regulates Rab11 and Endosomal Recycling"

_cells, 2020, doi:10.3390/cells9051132_

Round 1

Reviewer 1 Report

The paper by Wang et al. investigates the relation of the protein kalirin with Rab11. Kalirin has been previously described as a guanine exchage factor (GEF) for the RHO GTPase family involved in synapse formation and relevant for human diseases. In this manuscript the author discovered that Kalirin interacts with TRAPP, colocalizes with Rab11 on recycling endosomes and it regulates nucleotide exchange of Rab11, affecting the morphology recycling endosomes. The manuscript is interesting and it reports novel data that shed light on the function of Kalirin revealing that its activity can also affect Rab proteins and in particular Rab11. Thus they demonstrated that kalirin regulates both RHO and RAB GTPases.

Major points

-In Fig.1C while data on the 23-684 and 1269-1654 deletion constructs are convincing, the data on 674-1272 are not considering the low expression of this fragment. Indeed, considering the amount of the 23-684 fragment that binds to TRAPP compared to the loading amount, even if the 674-1272 was binding it would not show in the gel.

-In Fig. 3A the author show that Kalirin affect Rab11 nucleotide exchange but not Rab1.  I think that it will be important also to show the effect on a Rho protein to be able to compare the data.

-In Fig. 5A tubulation is not really visible so the authors should show images at higher magnification or resolution. Also it is hard to understand how from these images size of Rab11 structures has been measured.

Minor points

-In Fig.1C Ponseau should be Ponceau

-Fig.3B is not mentioned in the next.

-There are several types of small RNA (sRNA). The authors should indicate if they used siRNA or else and label figures accordingly.

Author Response

We appreciated this reviewer for suggestions and notifications of our errors.

Major points

-In Fig.1C while data on the 23-684 and 1269-1654 deletion constructs are convincing, the data on 674-1272 are not considering the low expression of this fragment. Indeed, considering the amount of the 23-684 fragment that binds to TRAPP compared to the loading amount, even if the 674-1272 was binding it would not show in the gel.

We noted this in Lines 91-92 of the Results. Since huntingtin interacts with Kalrn674-1272 (McClory et al. Sci Rep 2018) and with a Rab11GEF (Li et al NeuroReport 2008), we reasoned that the association of Kalrn674-1272 with trappc4 might result from the presence of huntingtin in the precipitates. We discussed this possibility in Lines 92-94 of the revised manuscript.

-In Fig. 3A the author show that Kalirin affect Rab11 nucleotide exchange but not Rab1.  I think that it will be important also to show the effect on a Rho protein to be able to compare the data.

Consistent with our previous report that kalirin immunoprecipitated from brain membranes had GEF activities toward Rac1 (McClory et al. Sci Rep 2018), immunoprecipitates obtained with anti-myc antibodies from cells expressing His-myc-kalirin promoted [3H]GDP release from Rac1. We included this data in revised fig. 3A.

-In Fig. 5A tubulation is not really visible so the authors should show images at higher magnification or resolution. Also it is hard to understand how from these images size of Rab11 structures has been measured.

As suggested, we showed images at higher magnifications in revised Fig. 5A. We described the analysis of Rab11 positive structures in the Methods and further clarified the definition of tubulated Rab11 endosomes as those Rab11 positive structures with a length of more than 3 folds of the width in Lines 453-454 of the Methods. 

Minor points

-In Fig.1C Ponseau should be Ponceau

We corrected this error in revised Fig. 1.

-Fig.3B is not mentioned in the next.

We corrected this in Line 167 of the Results.

-There are several types of small RNA (sRNA). The authors should indicate if they used siRNA or else and label figures accordingly.

We changed all “sRNA “into siRNA in the manuscript and figures and legends.

Reviewer 2 Report

Cells-776618 Kalirin interacts with TRAPP…..

Wang, …. Li

This potentially interesting manuscript is well written but in very important ways poorly reported, with crucial pieces of data missing. The basic idea that Kalirin interacts with the TRAPP complex of proteins in novel and potentially quite important and thus interesting, but the evidence is in many places incompletely presented. The proposal that the Kalirin-TRAPP complex plays a crucial role in long term potentiation is very important, if better documented and later corroborated by others. In addition, extremely standard and widely accepted terminology is not used, instead substituting the authors’ version of slightly different terminology, resulting in a quite confusing overall presentation.

Fig.1 co-ipt. This experiment lacks the usual controls, whether for the antibody pulldown or the immunoprecipitations. IgG controls are needed. In D, the EM image is very poor – a well-trained neuroscientist quite familiar with cortical immunoelectron microscopy would have no clue what the image shows – the authors unfortunately picked NRK cells instead of cortical tissue to image, without explanation. A and C simply must have molecular weight markers - the accepted standard is that no gel strip is so small that it cannot have at least 2 MW markers. C needs to have IgG controls. And while it is nice the “data are from one of three experiments”, the sentence needs to go on to say “with similar results” since the lack of conclusion raises the possibility that the 2 experiments not shown gave different results. Same comment for Fig.2.

The short RNAs (sRNAs) in Fig.2 are really siRNAs (synthetic 19 nt short interfering RNAs from Dharmacon) – there is absolutely no need to confuse/change the name. The well-experienced reader should be able to read the text and view the Figure and its legend without having to see the Methods section, and introducing novel nomenclature makes that impossible.

Similarly, it is confusing to substitute a unique abbreviation for Kalirin (Klrn) when the accepted gene nomenclature (Genbank, etc.) for two decades is one letter longer, Kalrn. Authors do themselves a disservice, since literature searches often use the gene name, Kalrn, and this article will be missed.

Figs.3-5 must have a clear description of how the imaging was controlled, with fixed exposure times.

Why switch red/green Kalirin/Rab11 in Fig.4? – this was a very unfortunate choice. And yes clearly colocalization is “rare” – but this almost looks like random luck! Lots and lots of dots in each color and occasionally they coincide, making it quite hard to get excited about the occasional coincidence. Same comment for Fig.5. **The really overriding question, which unfortunately would mean redoing the work in Figs. 4-5, is why the authors used NRK cells instead of brain sections. The staining and analysis would have been the same effort, but the biological impact of the answers would have been far greater.

Author Response

General comments on the use of NRK cells instead of brain tissues/neurons.

We appreciated this reviewer for the thoughtful comments and suggestions on the importance of the kalirin-TRAPP interaction in brain functions. Actually, we examined kalirin and Rab11/TRAPP co-localization initially in primary neurons, in which labeled structures were too crowded for us to make confident judgements. We therefore changed to use NRK cells for characterizing the new role for kalirin in regulating Rab11 and endosome trafficking because NRK cells are morphologically simpler than neurons to make it easier to distinguish individual organelles. It is necessary for future studies to investigate the dynamics and function of the kalirin-TRAPP complex in synaptic plasticity, particularly, in consideration that genetic studies have linked mutations in the Kalirin gene and genes of several TRAPP subunits (e.g., trappc2L, trappc4, trappc6a & trappc6b, trappc9, trappc11, and trappc12) to brain disorders.

Fig.1 co-ipt. This experiment lacks the usual controls, whether for the antibody pulldown or the immunoprecipitations. IgG controls are needed. A and C simply must have molecular weight markers - the accepted standard is that no gel strip is so small that it cannot have at least 2 MW markers. C needs to have IgG controls. And while it is nice the “data are from one of three experiments”, the sentence needs to go on to say “with similar results” since the lack of conclusion raises the possibility that the 2 experiments not shown gave different results. Same comment for Fig.2.

As suggested, we included the MW indications in revised Fig. 1, A and C. Data in Fig. 1C were obtained from GST pulldown experiments using GST-trappc4 proteins affinity purified from bacteria. Since GST alone did not pull down kalirin (Fig. 1A), we omitted the GST control in Fig. 1C. 

We added “with similar results” in Lines 114-115 and Line 145 of legends for Figs.1 and 2, respectively.

In D, the EM image is very poor – a well-trained neuroscientist quite familiar with cortical immunoelectron microscopy would have no clue what the image shows –.

The EM photograph showed a tubulovesicular structure co-labeled with trappc4 and kalirin. We revised Fig. 1D by adding two dashed red contours to trace the membranes of the tubulovesicular structure and indicated this revision in Line 123 of Fig. 1 legends.

The short RNAs (sRNAs) in Fig.2 are really siRNAs (synthetic 19 nt short interfering RNAs from Dharmacon) ….

We changed all “sRNA” into “siRNA”.

Similarly, it is confusing to substitute a unique abbreviation for Kalirin (Klrn) when the accepted gene nomenclature (Genbank, etc.) for two decades is one letter longer, Kalrn. Authors do themselves a disservice, since literature searches often use the gene name, Kalrn, and this article will be missed.

We changed all “Klrn” into “Kalrn”.

Figs.3-5 must have a clear description of how the imaging was controlled, with fixed exposure times.

For the comparison of signal intensities and endosome size, images in Figs. 3 and 5 were collected with the same settings including laser strength, pinhole, resolution, and scan times. We clarified these in Lines 444-446 of the Methods.

Why switch red/green Kalirin/Rab11 in Fig.4? – this was a very unfortunate choice. And yes clearly colocalization is “rare” – but this almost looks like random luck! Lots and lots of dots in each color and occasionally they coincide, making it quite hard to get excited about the occasional coincidence. Same comment for Fig.5. **The really overriding question, which unfortunately would mean redoing the work in Figs. 4-5, is why the authors used NRK cells instead of brain sections.

As this reviewer noted, the co-localization was very “rare”. The red/green switch for labeling Rab11 and Kalirin was a way we attempted to exclude the possible cross-reactivity of different sources of secondary antibodies, which might contribute to the “rare” co-localization events. We discussed the red/green switch in Lines 218-220 of the Results and revised Fig. 4B by adding a star symbol to indicate similar labeling patterns in a cell likely to be not transfected with plasmids expressing Rab11S25N.

We actually utilized primary neurons to examine kalirin and Rab11/TRAPP co-localization initially. However, labeled structures in neurons were too crowded for us to make confident judgements. Therefore, we used NRK cells for characterizing the new role of kalirin because they are morphologically simpler than neurons to make it easier to distinguish individual organelles.

Round 2

Reviewer 1 Report

I think the article is now acceptable for publication